# LSTM in Algorithmic Investment Strategies on BTC and S&P500 Index

**DOI:** 10.3390/s22030917

**Published:** 2022-01-25

**Authors:** Jakub Michańków, Paweł Sakowski, Robert Ślepaczuk

**Affiliations:** 1Doctoral School, Cracow University of Economics, ul. Rakowicka 27, 31-510 Cracow, Poland; jakub.michankow@phd.uek.krakow.pl; 2Quantitative Finance Research Group, Department of Quantitative Finance, Faculty of Economic Sciences, University of Warsaw, ul. Długa 44/50, 00-241 Warsaw, Poland; sakowski@wne.uw.edu.pl

**Keywords:** machine learning, recurrent neural networks, long short-term memory model, neural network, algorithmic investment strategies, systematic trading systems, loss function, walk-forward optimization, C4, C14, C45, C53, C58, G13

## Abstract

We use LSTM networks to forecast the value of the BTC and S&P500 index, using data from 2013 to the end of 2020, with the following frequencies: daily, 1 h, and 15 min data. We introduce our innovative loss function, which improves the usefulness of the forecasting ability of the LSTM model in algorithmic investment strategies. Based on the forecasts from the LSTM model we generate buy and sell investment signals, employ them in algorithmic investment strategies and create equity lines for our investment. For this purpose we use various combinations of LSTM models, optimized on in-sample period and tested on out-of-sample period, using rolling window approach. We pay special attention to data preprocessing in the input layer, to avoid overfitting in the estimation and optimization process, and assure correct selection of hyperparameters at the beginning of our tests. The next stage is devoted to the conjunction of signals from various frequencies into one ensemble model, and the selection of best combinations for the out-of-sample period, through optimization of the given criterion in a similar way as in the portfolio analysis. Finally, we perform a sensitivity analysis of the main parameters and hyperparameters of the model.

## 1. Introduction

The main aim of this paper is to explore deep learning possibilities in time series forecasting by applying buy/sell signals generated by the LSTM-type (Long Short-Term Memory) recurrent neural network to algorithmic investment strategies, tested on various frequencies of BTC (Bitcoin) and S&P500 index. We focus solely on LSTM networks, and compare its performance on various datasets, frequencies, selected hyperparameters and ensemble models, created by combining the aforementioned variables.

The main advantages and the novelty of our work can be divided into five important points, listed below. Firstly, the use of the newest Machine Learning (ML) methods (LSTM model) in algorithmic investment strategies (AIS) applied for cryptocurrency (BTC) and traditional equity index market (S&P500 index). Secondly, the indication of often encountered drawbacks occurring in paper testing of various algorithmic strategies. Thirdly, designing the proper architecture (initial hyperparameters tuning) of the LSTM model and testing the performance of AIS with comparison to the traditional Buy&Hold model (B&H). Fourthly, the use of various frequencies from daily to 15 min data in algorithmic investment strategies. Finally, the construction of an ensemble model, based on the combination of algorithmic investment strategies, on various frequencies applied for BTC and S&P500 index for separate and combined frequencies.

The idea for this paper arise from never ending attempts to understand and beat the market, through the construction of algorithmic investment strategies generating abnormal returns, i.e., characterized with risk adjusted returns significantly higher than the benchmark or other existing strategies. Moreover, none of the previous works covered the topic of the performance analysis of signals from the LSTM model in algorithmic investment strategies, with simultaneous focus on building a proper architecture of LSTM network, testing in on various frequencies and various asset classes with rolling window approach, enhanced with additional sensitivity analysis at the end. Although each year researchers publish multiple papers devoted to testing numerous alternative approaches employed in AIS, the results of these studies include numerous drawbacks and mistakes, which, in practice, makes it impossible to use in real trading. Therefore, the chase for an efficient algorithmic investment strategies continues.

The main research hypotheses verified in this paper are as follows:

**Hypothesis** **1**(**H1**)**.**
*The signals from LSTM model employed in AIS are more efficient than Buy&Hold approach, regardless of asset class tested.*

**Hypothesis** **2**(**H2**)**.**
*The signals from LSTM model employed in AIS are more efficient than Buy&Hold approach, regardless of data frequency tested.*

**Hypothesis** **3**(**H3**)**.**
*The signals from LSTM model employed in AIS are more efficient in case of BTC than in case of S&P500 index.*

**Hypothesis** **4**(**H4**)**.**
*The robustness of tested models to various hyperparameters does not depend on asset class tested.*

**Hypothesis** **5**(**H5**)**.**
*The ensemble model constructed as a combination of models with various frequencies and assets can produce better risk adjusted returns than single models.*

Referring to software, libraries, hardware and the time of calculation we can say that the results for LSTM model were obtained using R 4.1.0 along with Python 3.7.10. Deep learning libraries used for design, training and testing the network are Keras 2.4.0 and TensorFlow 2.5.0. The rest of the calculations, as well as graphs and tables were done using only R and RStudio environment. Computer specification used in this research was as follows: AMD Ryzen 7 3700X 3.6 GHz, 16 GB RAM, NVIDIA GeForce RTX 2060 Super with 270 tensor cores. One full training (number of rolling windows × 40 epochs) lasted around 20 min on 1 d frequency, 60 min on 1 h frequency, 180 min on 15 min frequency for S&P500 data, 80/240/720 min for BTC data.

The structure of this paper includes a short introduction with motivation and hypotheses in the Section 1 and literature review in the Section 2. Methodology and data is covered in the Section 3. The main results are presented in the Section 4. Then, the Section 5 covers the sensitivity analysis and Section 6 combined strategies, and the Section 7 concludes the research.

## 2. Literature Review

### 2.1. Most Common Drawbacks in Papers Testing AIS

Although literature review is very broad on this topic the main problem is that most of the papers testing algorithmic investment strategies do not maintain proper structure of testing, which is the reason why their results can not be treated as valid and robust. Before we go to the main empirical part of this research it is important to list and describe the most common drawbacks in papers testing various AIS, i.e.,:Only one in-sample and one out-of-sample period, causing that the results are heavily dependent on the selected period. This is a very common issue, that can be seen in majority of research papers on this topic (Wiecki et al. (2016) [1], Lopez de Prado (2013) [2], Bailey et al. (2016b) [3], Raudys (2016) [4]).Tests of AIS are performed on only one basis instrument, causing that results are strictly dependent on the characteristics of the distribution of this instrument (Vo and Yost-Bremm (2020) [5]).Over-optimization of ML models, (Lopez de Prado (2013) [2], Bailey et al. (2016b) [3]).Improper loss function or optimization criterion—in vast majority of the papers authors do not use proper optimization criteria (RMSE, MSE, MAE, MAPE %OP, etc.) and/or loss function, which makes it impossible to select the best techniques for creating buy/sell signals (Di Persio and Honchar (2016) [6], Yang et al. (2019) [7]).Forward looking bias in buy/sell signals—it is usually caused by the use of future macroeconomic data or errors in buy/sell signals definitions (Chan (2013) [8], Chan (2021) [9], Jansen (2020) [10]).No sensitivity analysis—such analysis enables authors to refer to the main results of the model, with regard to initially set parameters, and evaluate if they are robust (Di Persio and Honchar (2016) [6], Zhang et al. (2018a) [11], Yang et al. (2019) [7]).Data snooping bias—authors publish only the set of the best results obtained by the model, without consistent search among other of parameters and assumptions (Bailey et al. (2016a) [12], Chan (2013) [8]).Survivorship bias—the most common example of this bias is the selection of current constituents of the given index, in the research covering the last 20 years of data (Chan (2021) [9]).

### 2.2. LSTM Research Literature

Papers describing various approaches to LSTM can be diveded on those referring to the theoretical aspects of LSTM model and the ones focusing mainly on LSTM and various ML models empirical properties, tested on various sets of data.

The first introduction of LSTM was presented in the paper written by Hochreiter and Schmidhuber (1997) [13]. By introducing Constant Error Carousel (CEC) units, LSTM can deal with the exploding and vanishing gradient problems. The initial version of the LSTM block included cells, input, and output gates. LSTM genuine feature was the ability to preserve information through the chain of iterations during training. The next theoretical advancement was introduced by Gers (1999) [14] who introduced the forget gate (also called “keep gate”) into the LSTM architecture, enabling the network to reset its own state. Next, Gers et al. (2000) [15] added peephole connections, which are connections from the cell to the gates. Additionally, the output activation function was omitted. More recent advancements cover putting forward a simplified variant called Gated Recurrent Unit (GRU) by Chung et al. (2014) [16].

Another part of literature focuses on empirical research. Some studies provide either a general review of ML applications in financial time series forecasting (Heaton at al. (2016) [17], Tsantekidis et. al. (2017) [18], Rechentin (2014) [19]) or report performance of specific non-LSTM tools in this area (Tay and Cao (2002) [20], Sun et. al. (2017) [21], Van Gestel et. al. (2001) [22], Qu and Zhang (2016) [23]).

We can also find numerous studies presenting results of application the LSTM model, mostly in predicting stock prices.

Chen et al. (2015) [24] implemented LSTM on China stock market. They collected data from stocks and divided percentage returns of prices into seven groups: (−∞, −1.5], (−1.5, −0.5], (−0.5, 0.4], (0.4, 1.4], (1.4, 2.5], (2.5, 4.3], (4.3, ∞). The main aim of the research was tosuccessfully predict a proper group for the next day return. In addition to returns data, they also used 10 different features: open, low, high, close prices and volume for a given stock, and the same five features for Shanghai Securities Composite Index. Model specification used in this research: 30 days sequence length, ‘RMSprop’ optimizer and learning rate 0.001. The best results measured by accuracy of predicted return group were given by model using all ten features achieving 27.2% accuracy, which is twice as much as randomly picked groups. M’ng et al. (2016) [25] also explored trading with the FFN (Feedforward neural network) supported by the set of technical indicators (seven in total, downloaded from the Bloomberg website) as input variables. Their target was to predict close price changes of Kuala Lumpur Composite Index using historical data ranging from 2008 to 2014.

Roondiwala et al. (2017) [26] tried to predict stock returns of NIFTY 50 index. They trained multivariate LSTM model using daily OHLC prices as features. Model used 2 hidden layers with 128 and 64 units, each. Dense layer activation function was ‘ReLU’ and optimizer ‘RMSprop.’ Sequence length of days for individual input was 15 days. After testing different combinations of epochs and features the most accurate model in terms of RMSE used all four features and used 500 epochs for model training. Nelson et. al. (2017) [27], apply the LSTM model along with technical analysis indicators and get an average of 56% of accuracy in predicting directions of stocks movements in the near future. Bao et. al. (2017) [28] present a novel deep learning framework where wavelet transforms (WT), stacked autoencoders (SAEs) and long-short term memory (LSTM) are combined for stock price forecasting. Their model outperforms other similar models in both predictive accuracy and profitability performance.

Vargas et al. (2018) [29] used LSTM with input variables as technical indicators, but divided them into two sets. Both had a sequence length equal to period of 5. They also used the text analysis of financial news, which divided the study into the next two subgroups. Test was performed on the Chevron Corporation stocks between 2006 and 2013. The test period covered the last 8 months of the total set, which was equal to approximately 11% of data. The hyperparameters used in that work included: 128 LSTM units, 1 LSTM layer (LSTM as an input, no additional hidden layers), and SGD (stochastic gradient descent algorithm) as optimizer. The overall result exposed a great advantage of the LSTM network, supported by the news analysis and first set of TIs, over the standard LSTM model with the same set of TIs. Nevertheless, both of them proved to generate higher returns than buy-and-hold strategy.

Zhang et al. (2018b) [11] implemented LSTM model to predict the next day returns for China stocks. A different approach using LSTM model was presented in Sang and Di Pierro (2019) [30]. Instead of using prices or returns for predicting stock price movements, authors decided to use well known technical analysis trading strategies signals as features. Selected methods were: Simple Moving Average, Relative Strength Index and Moving Average Convergence Divergence. Dataset used in empirical study contained five stocks with highest capitalization in each from nine sectors of S&P500. Parameters used in final model were: one hidden layer, learning rate 0.001, 15 days sequence length. LSTM outperformed oscillators on six of nine sectors. Zhang et al. (2019) [31] presented AT-LSTM model which is combination of LSTM and Attention based model and provided results for three index datasets: Russell 2000, DJIA and NASDAQ. Kijewski and Ślepaczuk (2020) [32] compared the performance of classical techniques with LSTM model for S&P500 index on daily frequency from the last 20 years and showed that LSTM model results are not robust to initial hyperparameters assumptions.

One of the last approach which tested various machine learning techniques for time series forecasting problem was paper of Chlebus et al. (2020) [33] who applied the following methods: SVR, KNN, XGBoost, LightGBM, LSTM, ARIMA, ARIMAX with features coming from such classes like: technical analysis, fundamental analysis, Google Trends entries, markets related to Nvidia. The best performance was obtained by SVR based on stationary attributes.

At the end we have to say that many other authors have successfully verified that the LSTM network is able to perform better than many other popular time series prediction methods, examples include: (Gao and Chai (2018) [34], Dautel et al. (2020) [35], Fischer and Krauss (2018) [36], Shynkevich et al. (2017) [37]).

## 3. Methodology and Data

### 3.1. Terminology and Metrics

The main model used in this work is based on deep recurrent neural network, specifically on LSTM network. Performance of this type of networks proved to work very well with financial time series, and there have been extensive research put into testing LSTMs for stock returns forecasting and directional movements, as presented in literature review.

To train the network, a custom loss function had to be used as the base network performance metric in the training process. Apart from that, a set of strategy performance metrics was also calculated on the basis of equity line constructed from the investments based on single Buy/Sell signals. Sensitivity analysis was also implemented to show how changes in network hyperparameters and architecture affect the base case results. Additionally, ensemble models built on strategies with various frequencies and assets were tested at the end.

### 3.2. Lstm Model

LSTM networks are a type or recurrent neural networks (RNN) that can keep track of long term dependencies in data, allowing to partially solve vanishing gradient problem, typical for classic RNNs (Goodfellow et al. (2016) [38]). They are widely used to model sequential data such as text, speech and time series data. LSTM units are composed of memory cells, with each cell having three types of gates (input gate, output gate and forget gate). These gates use tanh and sigmoid functions to regulate the flow of information through the cell, deciding how much and which information should be stored in long term state, passed on to another step, or discarded. In our research, the input vector for the LSTM network ($x_{t}$), was a series of past observations form BTC and S&P500 data, and the output vector ($h_{t}$) was a single value predicted for the next period.

The architecture of LSTM network can be described as follows:(1)$$f_{t}=\sigma \left(U_{f}x_{t}+V_{f}h_{t-1}+b_{f}\right)$$
(2)$$C_{t}^{\prime }=f_{t}\circ C_{t-1}$$
(3)$$i_{t}=\sigma \left(U_{i}x_{t}+V_{i}h_{t˘1}+b_{i}\right)$$
(4)$$C_{t}^{+}=tanh\left(U_{c}x_{t}+V_{c}h_{t-1}+b_{c}\right)$$
(5)$$C_{t}=C_{t}^{\prime }+i_{t}\circ C_{t}^{+}$$
(6)$$o_{t}=\sigma \left(U_{o}x_{t}+V_{o}h_{t-1}+b_{o}\right)$$
(7)$$h_{t}=o_{t}\circ tanh\left(C_{t}\right)$$
where $f_{t}$, $i_{t}$ and $o_{t}$ are activation vectors for three specific gates, $h_{t}$ is a hidden state (or output) vector, $C_{t}$ is a cell state vector, while *b*, *U* and *V* denote biases, input weights and recurrent weights of the network cells. Figure 1 shows the a single cell of typical LSTM network:

### 3.3. Specification of Our LSTM Model

Our model consists of three LSTM layers with 512/256/128 neurons respectively and one single neuron dense layer on the output. Each of LSTM layers is using tanh activation function, which allows to retain negative values. L2 kernel regularization (0.0005) and dropout (0.02) are also applied to each of these layers. Input shape of the data for the network was set to (sequence size, number of features), where only one feature was used as the input data—the simple returns of the tested frequency. The first two layers return sequences with the same shape as the input sequence (full sequence) and the last LSTM layer returns only the last output. To be able to use GPU acceleration during the training process, recurrent activation function was set to sigmoid and we did not use any recurrent dropout.

To train the model, we used Adam optimizer (Kingma and Ba (2017) [40])—a stochastic gradient descent optimizer with momentum (estimating first-order and second-order moments). The learning rate of the optimizer was set to 0.0015 (after tuning). Data was split into mini-batches (set to 80 after tuning), to allow the optimizer to work more efficiently. Such architecture allowed us to use the model efficiently across both datasets, as well as test it on different frequencies and apply sensitivity analysis to various hyperparameter settings.

#### 3.3.1. New Loss Function

In order to avoid one of the most common drawbacks from papers testing AIS, we introduced our authorship loss function, which improves the usefulness of forecasting ability of LSTM model in algorithmic investment strategies (AIS).

Based on our previous research (e.g., Kijewski and Ślepaczuk (2020) [32]), and Vo and Ślepaczuk (2022) [41], we concluded that popular error metrics like RMSE, MSE, MAE, MAPE, %OP used in 99.9% of similar research are not proper error functions for the evaluating the efficiency of forecasting ability of the models tested in AIS. The reason is that the above mentioned error metrics evaluate only the accuracy of forecasts (i.e., difference between forecasted and observed value), which is often confused with the forecasting ability of investment signals in AIS built on these forecasts. It means that almost all these error metrics (RMSE, MSE, MAE, MAPE) are penalized no matter if the forecast error ($\mathrm{forecast}\mathrm{error}={\hat{R}}_{i}-R_{i}$) was positive or negative while %OP metric does not take into account the magnitude of forecast error, but only its direction. For this reason, researchers in most of the other papers select not the most profitable combination of signals for the strategy, but the combination which only optimizes the selected error metric.

Therefore, we propose new loss function called Mean Absolute Directional Loss (MADL) that can be calculated using the following formula:(8)$$\mathrm{MADL}=\frac{1}{N}\sum_{i=1}^{N}\left(-1\right)\times \mathrm{sign}\left(R_{i}\times {\hat{R}}_{i}\right)\times \mathrm{abs}\left(R_{i}\right)$$
where: MADL is the Mean Absolute Directional Loss, $R_{i}$ is the observed return on interval *i*, ${\hat{R}}_{i}$ is the predicted return on interval *i*, sign{X} is the function which gives the sign of *X*, abs{X} is the function which gives the absolute value of *X* and *N* is the number of forecasts. This way, the value the function returns will be equal to the observed return on investment with the predicted direction, which allows the model to tell if the prediction will yield profit or loss and how much this profit or loss will be. MADL was designed specifically for working with AIS’s. The function in our model is minimized, so that if it returns the negative values the strategy will make a profit, and if it returns a positive value the strategy will generate a loss.

MADL was the main loss function used in hyperparameters tuning and in the estimation of the LSTM model.

### 3.4. Hyperparameters Turning

During our research we conducted detailed hyperparameters tuning to ensure the best possible results from our model. During the process we optimized the following parameters:The number of layers (1–5) and neurons in each layer (5–512).Dropout rate (0.001–0.2) and l2 kernel regularization (0.0001–0.01).Several different types of the optimizer (SGD, RMSProp and Adam variants).Learning rates (0.001–0.1) and momentum values (0.1–0.9).

As for the input data, we tuned the training and testing/rolling window sizes, sequence length (2–20), batch size (from 16 to test size) and training process duration, which was set by the number of epochs (10–300), as well as callbacks functions of early stopping and model checkpoint. Only the first window of data was used for tuning, and the best hyperparameters were then used for the remaining iteration during the walk forward predictions.

Most of the tuning was done using the KerasTuner framework (O’Malley et al. (2019) [42]) allowing automated parameter selection using Hyperband search algorithm (Li et al. (2018) [43]). This approach allowed us to test how changes to several parameters at once would affect the network performance, instead of testing each hyperparameter separately. Results are presented in Table 1. In addition, we also conducted a careful manual sensitivity analysis on the parameters that had the most impact on the results.

### 3.5. Training Process

For training and prediction we used a walk forward predictions/rolling window approach. This allowed us to make sure that the network will not overfit, as it was trained and tested multiple times, across various sets of data. Model was trained on approximately three years of data (equal to train set length) and then it was used for predictions over the next 3 months (equal to test set length). During that period, a single return value was predicted each time, based on the last 14/20 (sequence length) values. After making the predictions, the window was moved ahead, by the number of periods equal to test set and the model was retrained from scratch.

A single iteration was trained for 40 epochs. Model checkpoint callback function was used to store the best weights (parameters) of the model, based on the lowest loss function value from all trained epochs. These weights were then used for prediction on the test set data.

### 3.6. Research Description

During this research the following steps were performed:The division into in-sample (training and validation) and into out-of-sample (test) samples, set to 1371/90 observations for BTC and 948/65 observations for S&P500.Hyperparameters tuning, described in Section 3.4.Buy/Sell signals definitions based on the next day forecasts.Tests for two types of strategies: Long/Short and Long only.New Loss function: MADL, described in Section 3.3.1.Walk-forward predictions.Equity lines and performance metrics according to Ślepaczuk et al. (2018) [44], results provided in Section 4.Sensitivity analysis for various values of Dropout, Sequence length, Train Set length, Batch size, results provided in Section 5.The combination of signals across different frequencies (1 d, 1 h and 15 min) and asset classes (equity—S&P500 index and cryptocurrency—BTC), results provided in Section 6.

### 3.7. Performance Metrics

In order to evaluate the efficiency of tested strategies we calculate the following performance metrics based on Kosc et al. (2019) [45] and Bui and Ślepaczuk (2021) [46].Annualized Return Compounded (ARC), which shows annualized rate of return for the given instrument (strategy), over the period (0,…,T):
(9)$$\mathrm{ARC}=({\left(1+\frac{P_{T}}{P_{0}}\right)}^{S/T})\times 100\%$$
where $P_{T}$ is the price of the given instrument at the end of interval *T*, $P_{0}$ is its current price and the scale parameter *S* is equal to the number of trading periods during a year for a given frequency.Annualized Standard Deviation (ASD) is the most common risk measure showing the annualized deviation of returns from their long-term average:
(10)$$\mathrm{ASD}=\sqrt{\frac{S}{T}\sum_{k=0}^{T}{\left(R_{t-k}-\bar{R}\right)}^{2}}\times 100\%$$
where $\bar{R}$ is the average simple daily return of the given instrument and the scale parameter *S* is equal to the number of trading periods during a year for a given frequency.Maximum Drawdown (MD) which informs us about maximum percentage drawdown during the investment period:
(11)$$\mathrm{MD}\left(T\right)=\max_{\tau \in [0,T]}\left(\max_{t\in [0,\tau ]}\left(R_{i,T}-R_{i,\tau }\right)\right)\times 100\%$$Maximum Loss Duration (MLD) which informs us about maximum number of years between the previous local maximum to the forthcoming local maximum:
(12)$$\mathrm{MLD}=\max\frac{m_{j}-m_{i}}{S}$$
for which ${\mathrm{Val}}_{m_{j}}>{\mathrm{Val}}_{m_{i}}$ and j>i, where $m_{j}$ and $m_{i}$ are the numbers of days indicating consecutive local maximum of equity line, ${\mathrm{Val}}_{m_{j}}$ and ${\mathrm{Val}}_{m_{i}}$ are the values of local maximums in days $m_{j}$ and $m_{i}$, respectively. The scale parameter *S* is equal to the number of trading periods during a year for a given frequency.Information Ratio (IR^*^) which describes the relation of the portfolio annualized rate of return to its annualized standard deviation:
(13)$${\mathrm{IR}}^{*}=\frac{\mathrm{ARC}}{\mathrm{ASD}}$$Modified Information Ratio (IR^**^) which takes into account the sign of the ARC metric:
(14)$${\mathrm{IR}}^{**}={\mathrm{IR}}^{*}\times \mathrm{ARC}\times \frac{\mathrm{sign}(\mathrm{ARC})}{\mathrm{MD}}$$Aggregated Information Ratio (IR^***^) which we regard this as the most important in evaluation of results of this study:
(15)$${\mathrm{IR}}^{***}=\frac{{\mathrm{ARC}}^{3}}{\mathrm{ASD}\times \mathrm{MD}\times \mathrm{MLD}}$$Number of observations (nObs) which is the length of the investment horizon in trading days.Number of trades (nTrades).

### 3.8. Data Description

As input data for the network we used simple returns, based on one minute data for both Bitcoin (BTC) and S&P500, from 1 April 2013 to 31 December 2020 (source for the BTC data: Kraken, Bitfinex, BTC-e, CEX and Coinbase exchanges. Source for the S&P500 data: Interactive Brokers API). Lower frequency returns were aggregated from the minute returns data. The descriptive statistics for time series on daily, hourly and 15 min frequency are presented in Table 2.

For the size of the training set we used 1371 observations for BTC and 948 observations for S&P500, (after tuning). Validation set size was 33% of the training set. Test sets, and also rolling window, size was set to 90 for BTC and 65 observations for S&P500 (after turning).

Input sequence size for LSTM network was set to 20 for BTC and 14 for S&P500 and the batch size was set to 80.

The output of the model was a single number predicting the next return value. Based on the sign of the predicted return value we assigned −1, 0, 1 signals. However, the cases where network predicted the 0 return value (resulting in a neutral signal) were negligible.

The hours of trading for S&P500 index were between 3.30 p.m. CET and 10.00 p.m. CET, from Monday to Friday excluding official holidays, while BTC was traded 24 h per day, 7 days a week.

## 4. Results for the Base Case Scenario

Table 3 shows the results of strategies for BTC and S&P500 on daily frequency, using a classic MSE loss function and our novel MADL loss function. MSE was used as our starting point in comparing the performance of LSTM networks but after thorough consideration placed in Section 3.3.1 and comparison placed in Table 3 we saw that results based on MADL function are much better in terms of maximizing of our risk-adjusted return metrics (IR^*^, IR^**^, and IR^***^) for tested algorithmic investment strategies. Therefore, in the next steps, we use our novel MADL function.

Table 4 presents the aggregated results for all frequencies for BTC and S&P500 index. Panel A in Table 4 shows that the best results for BTC on daily frequency with regards to aggregated IR were obtained by Long/Short strategy (IR^***^ = 21.23), but Long Only had very similar results (IR^***^ = 19.56), which were much better than for Buy&Hold strategy on BTC (IR^***^ = 1.19). The best results for Long/Short and Long Only strategies were possible mainly because the joint improvement of ARC, MD, and MLD indicators.

Panel B in Table 4 shows that the best results for BTC on hourly frequency, with regards to aggregated IR, were obtained by Long Only strategy (IR^***^ = 7.12), while Long/Short and BTC had much worse results (IR^***^ = 0.49) and (IR^***^ = 1.13).

Panel C in Table 4 shows that the best results for BTC on 15 min frequency, with regards to aggregated IR, were obtained by BTC strategy (IR^***^ = 1.11) while Long/Short and Long Only had much worse results (IR^***^ = −0.05) and (IR^***^ = 0.05).

Panel D, E and F in Table 4 summarize the results for strategies on S&P500 index. Panel D for daily frequency shows that the best results were obtained for S&P500 and Long Only (IR^***^ = 0.09 and IR^***^ = 0.09). Panel E for hourly frequency shows that the best results were obtained for S&P500 (IR^***^ = 0.05). Panel F for 15 min frequency shows that the best results were obtained for Long Only (IR^***^ = 0.38).

Figure 2 presents the equity lines for all frequencies for BTC and S&P500 index. Panel A on Figure 2 shows the fluctuations of equity lines for tested strategies, and confirms the results presented in Table 1. The equity lines for Long/Short and Long Only strategies climb higher in a much smoother way than for BTC. The similar confirmation can be seen in Panels B and C of Figure 2 for BTC and in Panels D, E and F of Figure 2 for S&P500 index.

Table 5 shows the results of a test of significance of α and β from the regression in the form of $R_{t}=\alpha +\beta R_{t}^{*}+\varepsilon_{t}$, where $R_{t}$ is the buy and hold returns, and $R_{t}^{*}$ returns from Long/Short and Long Only strategies, and test whether α=0 using standard tools and additionally one paragraph with the interpretation of results. Generally, the results presented in Table 5 confirm the ones presented in Table 4, while the slight differences come from different approach to risk metrics, mainly MD and MLD.

Summarizing the results for investment strategies on BTC in the base case scenario, we should underline that on the daily frequency the best results were obtained for Long/Short and Long Only, on hourly frequency for Long Only, and 15 min frequency for BTC. Therefore, we can note that in case of BTC the results worsen when we change the frequency from daily to hourly and then to 15 min.

Slightly different situation can be observed for investment strategies on S&P500 index in the base case scenario. We can notice that on the daily frequency the best results were obtained for S&P500 and Long Only, on hourly frequency for S&P500, and 15 min frequency for Long Only. Overall, we can note that in case of S&P500 index the best results were for 15 min, then for 1 d and lastly for 1 h. Additionally, we see that LO is much better than LS.

## 5. Sensitivity Analysis

In order to properly refer to obtained results, we have to check their robustness with regards to all crucial hyperparameters that were selected at the beginning. Therefore, in this section we checked the sensitivity of results based on the changes to the following hyperparameters: dropout rate, sequence length, train set length and batch size, changing them one by one, *ceteris paribus*. Due to the very long time of computations, we were not able to perform the analysis for all the frequencies (especially for 15 min data). Therefore we decided to present it only for hourly frequency. We check the sensitivity of final investment strategies to the changes of the following values of above mentioned hyperparameters:Dropout rate: from 2% to 1% and 4%.Sequence length.–For BTC: from 20 to 10 and 40.–For S&P500: from 14 to 7 and 28.Train set length.–For BTC: from 1371 to 685 and 2742.–For S&P500: from 948 to 474 and 1896.Batch size: from 80 to 40 and 160.

### 5.1. Sensitivity Analysis for 1 h Data—Dropout Rate

Table 6 presents the aggregated results of sensitivity analysis for 1 h data for BTC and S&P500 for Long/Short and Long Only strategies.

The short summary of the sensitivity of tested strategies to the changes in dropout rate is listed below. In case of Long/Short for BTC, the most efficient dropout was 2%, i.e., the one selected during hyperparameters tuning. The results of the model are not robust to slight changes in dropout rate what can be additionally seen in Panel A of Figure 3.

The most efficient dropout in case of Long Only for BTC was 2%, i.e., once again the one selected during hyperparameters tuning. The results of the model are rather robust to slight changes in dropout rate (Panel B of Figure 3).

The results of sensitivity analysis to the changes in dropout rate differ when we take into account the S&P500 index. The most efficient dropout in case of Long/Short strategy was 1%, while 2% selected during hyperparameters tuning was the least efficient. The results of the model were quite robust to slight changes in dropout rate,

In the case of Long Only for S&P500 index, the most efficient dropout was 1%, but 2% selected during hyperparameters tuning gives almost the same results. The results of the model are robust to slight changes in dropout rate.

Summarizing sensitivity of tested strategies to the changes in dropout rate we can say that strategies for BTC were not robust, while they were robust for S&P500 index. Moreover, we noticed that the parameters selected during hyperparameters tuning were still the best after sensitivity analysis in case of BTC strategies and it was not the case for S&P500 index.

### 5.2. Sensitivity Analysis for 1 h Data—Sequence Length

Details of sensitivity analysis to the changes in sequence length are presented in Table 7 and Figure 4.

Panel A of Table 7 and Figure 4 show that in case of Long/Short for BTC the most efficient sequence length was 20, i.e., the one selected during hyperparameters tuning and that the results of the model were not robust to slight changes in sequence length.

In the case of a Long Only strategy for BTC (Panel B of Table 7 and Figure 4 the most efficient sequence length was 20, i.e., the one selected during hyperparameters tuning and the results of the model were not robust to slight changes in sequence length.

The results for S&P500 index were slightly different because the sensitivity analysis showed that the best sequence length for Long/Short (Panel C of Table 7 and Figure 4 and Long Only (Panel D of Table 7 and Figure 4 was 7, while 14 selected during hyperparameters tuning was the least efficient. Moreover, the results of the model were not robust to slight changes in dropout rate.

Summarizing the results of sensitivity analysis to the changes in Sequence length we can say that strategies for BTC nor for S&P500 index were not robust. Moreover, we noticed that the parameters selected during hyperparameters tuning were still the best after sensitivity analysis in case of BTC strategies and it was not the case for S&P500 index.

### 5.3. Sensitivity Analysis for 1 h Data—Train Set Length

Table 8 and Figure 5 show the results of sensitivity analysis to the changes in train set length.

Panel A of Table 8 and Figure 5 presenting the results for Long/Short strategy for BTC and panel B of Table 8 and Figure 5 presenting the results for Long Only strategy for BTC informs us that the most efficient Train Set length was 1371, i.e., the one selected during hyperparameters tuning. The results of the model were not robust even to slight changes in sequence length.

The results for S&P500 index are once again slightly different. In case of Long/Short (Panel C of Table 8 and Figure 5) and Long Only (Panel D of Table 8 and Figure 5) the most efficient train set length was 1896, while 948 selected during hyperparameters tuning was the least efficient. The results of the model were not robust to slight changes in train set length.

Summarizing the results of sensitivity analysis to the changes in Train set length we can say that strategies for BTC nor for S&P500 index were not robust. Moreover, we noticed that the parameters selected during hyperparameters tuning were still the best after sensitivity analysis in case of BTC strategies and it was not the case for S&P500 index.

### 5.4. Sensitivity Analysis for 1 h Data—Batch Size

The last part of sensitivity analysis is summarized in Table 9 and Figure 6.

The results of sensitivity analysis to the changes in Batch size for BTC for Long/Short (Panel A of Table 9 and Figure 6) and Long Only (Panel B of Table 9 and Figure 6) show that the most efficient Batch Size was 80, i.e., the one selected during hyperparameters tuning. The results of the model were not robust to slight changes in batch size.

The results of sensitivity analysis to the changes in Batch size for S&P500 index for Long/Short (Panel C of Table 9 and Figure 6) and Long Only (Panel D of Table 9 and Figure 6) show that the most efficient Batch Size was 40, while 80 selected during hyperparameters tuning was the least efficient. The results of the model were not robust to slight changes in batch size.

Summarizing the results of sensitivity analysis to the changes in Batch size we can say that strategies for BTC nor for S&P500 index were not robust. Moreover, we noticed that the parameters selected during hyperparameters tuning were still the best after sensitivity analysis in case of BTC strategies and it was not the case for S&P500 index.

Overall the results of sensitivity analysis inform us that it is possible that hyperparameters tuning procedure was correct for BTC but should be improved for S&P500.

## 6. Combined Model on Different Frequencies and Different Assets

In order to smooth our equity lines and use limited correlations between AIS on various frequencies and various types of assets, we decided to create ensemble models built from three frequencies (1 d, 1, and 15 min) and/or two types of assets (BTC and S&P500).

We have tested two ways of combinations of signals across frequencies used(1 d, 1 h and 15 min):Approach #1: three signals {1, −1, 1} in the same interval are combined as {1/3}.Approach #2: three signals {1, −1, 1} in the same interval are combined as {1}.

However, due to very similar results, we have decided to present the results only for approach #1. Other aspects of construction of equity lines stays as it was for the base case in the main results section.

Table 10 and Figure 7 summarize the result of ensemble models for BTC and S&P500 index. Panel A of Table 10 with ensemble model for combined frequencies on BTC, shows that the most efficient results can be obtained for Long Only strategies and for BTC strategy. The similar results are presented in Panel B of Table 10 for S&P500 index. Overall, we can notice that models ensembled across frequencies results in lower volatility and smoother equity lines.

Summarizing the results for ensemble model built on different frequencies and different assets presented in Panel C, D, E, and F of Table 10 we can stress the following:The combination of weights equal to {S&P500 = W20%, BTC = 80%} was always better than {S&P500 = W10%, BTC = W90%}.The length of rebalancing period equal to RB6m was always better than RB3m.

Panels A, B, C, D, E, and F of Figure 7 confirm the observations from Table 10.

As a conclusion for this section, we can say that combined results for ensemble model suggest rare rebalancing of assets and higher weight of BTC in the optimal portfolio for investment strategies.

## 7. Conclusions

This research aimed to test LSTM networks in forecasting the value of the BTC and S&P 500 index on the data from 2013 to the end of 2020 on data with the following frequencies: daily, 1 h and 15 min. Based on the forecasts from LSTM models we generated buy and sell investment signals, used them in algorithmic investment strategies and created equity lines for our investment. For this purpose we used various combination of LSTM models optimized on in-sample period and tested on out-of-sample period with rolling window approach. We paid special attention to data preprocessing in the input layer, to avoiding overfitting in the estimation and optimization process, and we assured correct selection of hyperparameters at the beginning of our tests. We introduced our authorship loss function with better utilizes the forecasting ability of LSTM model in algorithmic investment strategies. Then we performed the sensitivity analysis of the main parameters and hyperparameters. In the final step, we combined the signals from various frequencies into one ensemble model.

In this paragraph we refer to Research Hypotheses formulated at the beginning of this research.

The first hypothesis (H1): *The signals from LSTM model employed in AIS are more efficient than Buy&Hold approach regardless of asset class tested*,holds only for some of BTC strategies (1 d_LS, 1 d_LO, and 1 h_LO) and some of S&P500 strategies (1 d_LO, 15 min_LS, and 15 min_LO). Therefore, we reject H1.

The second hypothesis (H2): *The signals from LSTM model employed in AIS are more efficient than Buy&Hold approach regardless of data frequency tested*, holds only for daily data in case of BTC and 15 min data in case of S&P500. Therefore, we reject H2.

The third hypothesis (H3): *The signals from LSTM model employed in AIS are more efficient in case of BTC than in case of S&P500 index*, has to be rejected as well because for various frequencies we obtain different results.

The fourth hypothesis (H4): *The robustness of tested models to various hyperparameters does not depend on asset class tested*, enables us to state that the results were not robust for BTC nor for S&P500 but in significantly different way. Nevertheless, we can not reject the fourth hypothesis.

Finally, we were not able to reject the fifth hypothesis (H5): *The ensemble model constructed as a combination of ML models with various frequencies and assets can produce better risk adjusted returns than single models*, because Long only ensemble strategies performed best.

Summarizing the most important conclusions from this paper, we can state that the efficiency of LSTM in algorithmic investment strategies strictly depends on the hyperparameters tuning procedure, the construction of the model and the estimation process. Moreover, the proper loss function is crucial in the model estimation process. What is more, the results are dependent on asset classes and frequencies used. Finally, we noticed that the results are not robust to initial assumptions.

Possible research extensions of this paper could cover more extensive sensitivity analysis especially with regards to parameters and hyperparameters which were not tested in this study, the construction of alternative loss functions improving the problems identified with regards to common error measures (one of the drawbacks of using MADL as a loss function is that it’s not easily optimized), the use of various high frequency data, the repetition of the whole research with transaction costs included in the estimation process and finally, more careful hyperparameters tuning process, especially in case of S&P500.

## Figures and Tables

**Figure 1 sensors-22-00917-f001:**
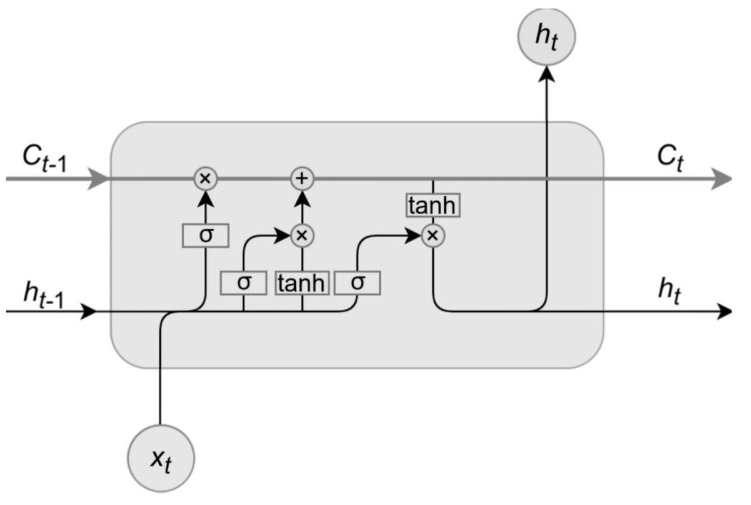
Architecture of LSTM cell. Source: Matsumoto F., (2019) [39].

**Figure 2 sensors-22-00917-f002:**
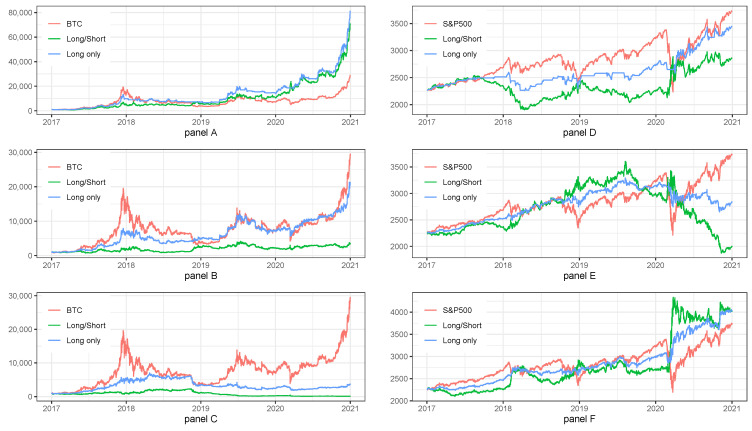
Equity lines for investment strategies in BTC and S&P500 in the base case scenario, compared with benchmarks. Note: BTC and S&P500 stand for the benchmark strategies, i.e., Buy&Hold applied for BTC and S&P500 prices, respectively. Long/Short stands for the investment strategy with long and short signals. Long only stands for the investment strategy with long only signals. The plot presents the fluctuations of equity lines in the period between 1 January 2017 and 31 December 2020 for daily frequency. Panel (**A**) and panel (**B**) shows the results for daily frequency for BTC and S&P500 index, respectively. Panel (**C**) and panel (**D**) shows the results for hourly frequency for BTC and S&P500 index, respectively. Panel (**E**) and panel (**F**) shows the results for 15 min frequency for BTC and S&P500 index, respectively.

**Figure 3 sensors-22-00917-f003:**
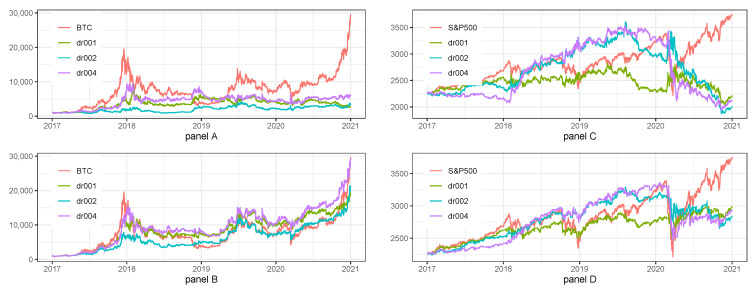
Sensitivity analysis for 1 h data with regard to the dropout rate. Note: BTC and S&P500 stand for the benchmark strategies, i.e., Buy&Hold applied for BTC and S&P500 prices, respectively. dr001, dr002, dr004 is the abbreviation for dropout rates equal to 1%, 2%, and 4%. The plot presents the fluctuations of equity lines in the period between 1 January 2017 and 31 December 2020 for daily frequency. The hyperparameters of LSTM model for the base case scenario were set as it was described in Table 1. Panel (**A**) presents the results for Long/Short strategies on BTC. Panel (**B**) presents the results for Long only strategies on BTC. Panel (**C**) presents the results for Long/Short strategies on S&P500 index. Panel (**D**) presents the results for Long only strategies on S&P500 index.

**Figure 4 sensors-22-00917-f004:**
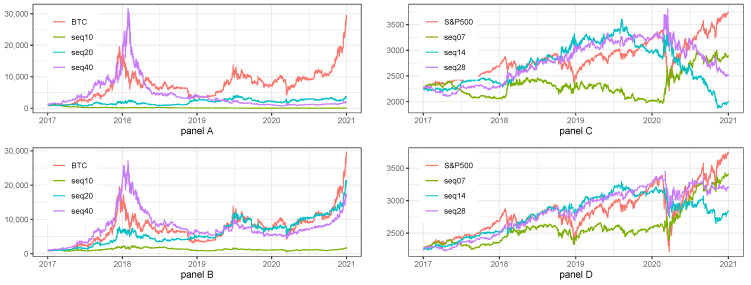
Sensitivity analysis for 1 h data with regard to sequence length. Note: BTC and S&P500 stand for the benchmark strategies, i.e., Buy&Hold applied for BTC and S&P500 prices, respectively. seq10, seq20, seq40 is the abbreviation for sequence lengths equal to 10, 20, and 40. The plot presents the fluctuations of equity lines in the period between 1 January 2017 and 31 December 2020 for daily frequency. The hyperparameters of LSTM model for the base case scenario were set as it was described in Table 1. Panel (**A**) presents the results for Long/Short strategies on BTC. Panel (**B**) presents the results for Long only strategies on BTC. Panel (**C**) presents the results for Long/Short strategies on S&P500 index. Panel (**D**) presents the results for Long only strategies on S&P500 index.

**Figure 5 sensors-22-00917-f005:**
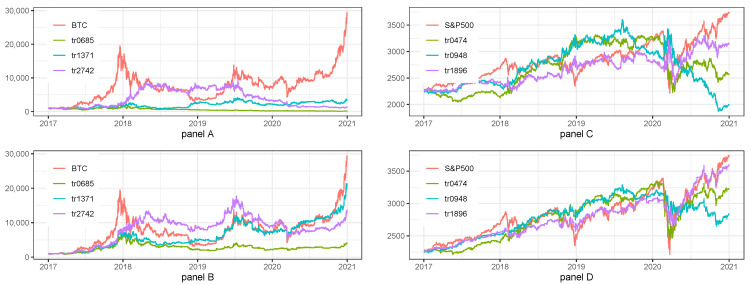
Sensitivity analysis for 1 h data with regard to the train set length. Note: BTC and S&P500 stand for the benchmark strategies, i.e., Buy&Hold applied for BTC and S&P500 prices, respectively. tr0685, tr1371, tr2742, tr0474, tr0948, tr1896 is the abbreviation for train set lengths equal to 685, 1371, 2742, 474, 948 and 1896. The plot presents the fluctuations of equity lines in the period between 1 January 2017 and 31 December 2020 for daily frequency. The hyperparameters of LSTM model for the base case scenario were set as it was described in Table 1. Panel (**A**) presents the results for Long/Short strategies on BTC. Panel (**B**) presents the results for Long only strategies on BTC. Panel (**C**) presents the results for Long/Short strategies on S&P500 index. Panel (**D**) presents the results for Long only strategies on S&P500 index.

**Figure 6 sensors-22-00917-f006:**
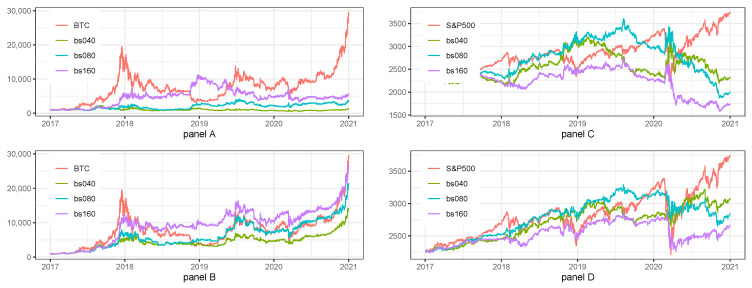
Sensitivity analysis for 1 h data with regard to the batch size. Note: BTC and S&P500 stand for the benchmark strategies, i.e., Buy&Hold applied for BTC and S&P500 prices, respectively. bs040, bs080, bs160 is the abbreviation for batch sizes equal to 40, 80 and 160. The plot presents the fluctuations of equity lines in the period between 1 January 2017 and 31 December 2020 for daily frequency. The hyperparameters of LSTM model for the base case scenario were set as it was described in Table 1. Panel (**A**) presents the results for Long/Short strategies on BTC. Panel (**B**) presents the results for Long only strategies on BTC. Panel (**C**) presents the results for Long/Short strategies on S&P500 index. Panel (**D**) presents the results for Long only strategies on S&P500 index.

**Figure 7 sensors-22-00917-f007:**
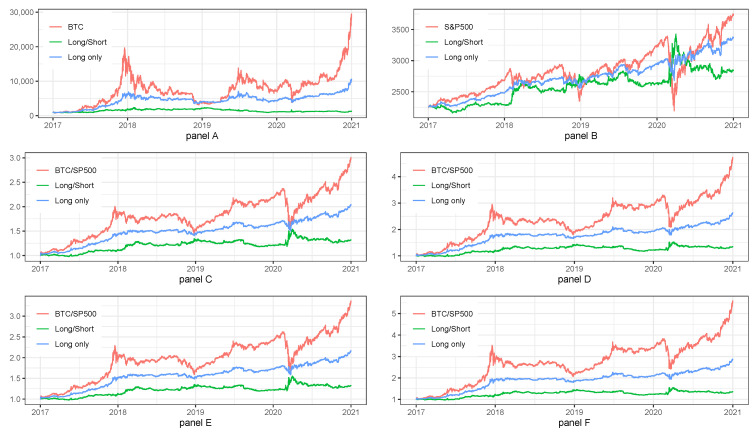
Combined model on different frequencies and different assets. Note: BTC and S&P500 stands for the benchmark strategy, i.e., Buy&Hold applied for BTC price and S&P500 index. Long/Short stands for the investment strategy with long and short signals. Long only stands for the investment strategy with long only signals. The plot presents the fluctuations of equity lines in the period between 1 January 2017 and 31 December 2020 for daily frequency. Panel (**A**) shows the results of ensemble model built on three various frequencies for BTC. Panel (**B**) shows the results of ensemble model built on three various frequencies for S&P500 index. Panel (**C**) shows the results of ensemble model built on three various frequencies for BTC and S&P500 with rebalancing period equal to 3m and weights equal to 10% for BTC and 90% for S&P500 index. Panel (**D**) shows the results of ensemble model built on three various frequencies for BTC and S&P500 with rebalancing period equal to 3m and weights equal to 20% for BTC and 80% for S&P500 index. Panel (**E**) shows the results of ensemble model built on three various frequencies for BTC and S&P500 with rebalancing period equal to 6m and weights equal to 10% for BTC and 90% for S&P500 index. Panel (**F**) shows the results of ensemble model built on three various frequencies for BTC and S&P500 with rebalancing period equal to 6m and weights equal to 20% for BTC and 80% for S&P500 index.

**Table 1 sensors-22-00917-t001:** Values of hyperparameters selected after network tuning.

Hyperparameter	Selected Value
No. hidden layers	3
No. neurons	512/256/128
Activation function	tanh
Dropout rate	0.02
l2 regularizer	0.0005
Optimizer	Adam
Learning rate	0.00015
BTC train/test	1371/90
S&P train/test	948/65
Batch size	80
Sequence length	14/20

Source: Own study.

**Table 2 sensors-22-00917-t002:** Descriptive statistics for BTC and S&P500 returns.

	Daily	Hourly	15 min
**panel A: BTC**			
Min.	−0.572057	−0.1790734	−1.321 $\times \;10^{-1}$
1st Qu.	−0.013333	−0.0027008	−1.507 $\times \;10^{-3}$
Median	0.001667	0.0001258	3.835 $\times \;10^{-5}$
Mean	0.003911	0.0001444	3.683 $\times \;10^{-5}$
3rd Qu.	0.019457	0.0030950	1.648 $\times \;10^{-3}$
Max	3.368390	0.2088179	1.915 $\times \;10^{-1}$
Skew.	24.92251	0.02863625	0.1060641
Kurt.	1002.3	31.59648	62.33809
Norm.	<2.2 $\times \;10^{-16}$	<2.2 $\times \;10^{-16}$	<2.2 $\times \;10^{-16}$
**panel B: S&P500**			
Min.	−0.1202281	−9.533 $\times \;10^{-2}$	−8.124 $\times \;10^{-1}$
1st Qu.	−0.0032111	−1.059 $\times \;10^{-3}$	−5.424 $\times \;10^{-4}$
Median	0.0006892	1.402 $\times \;10^{-4}$	4.214 $\times \;10^{-5}$
Mean	0.0005085	8.239 $\times \;10^{-5}$	2.340 $\times \;10^{-5}$
3rd Qu.	0.0051177	1.367 $\times \;10^{-3}$	6.035 $\times \;10^{-4}$
Max	0.0943437	5.680 $\times \;10^{-2}$	5.309 $\times \;10^{-2}$
Skew.	−0.6935947	−1.575707	−3.019853
Kurt.	21.11677	61.30094	173.2111
Norm.	<2.2 $\times \;10^{-16}$	<2.2 $\times \;10^{-16}$	<2.2 $\times \;10^{-16}$

Note: These descriptive statistics were calculated for each time frequency separately in the period staring 1 April 2013 for daily frequency for both BTC and S&P500, BTC hourly data from 4 November 2016, 15 min data from 17 December 2016, S&P500 hourly data from 20 June 2016, 15 min data from 11 August 2016 and all sets finishing on 31 December 2020. Norm. is Pearson chi-square normality test *p*-value.

**Table 3 sensors-22-00917-t003:** Performance metrics for investment strategies on BTC and S&P500 in the base case scenario with MSE and MADL loss functions.

	aRC	aSD	MD	MLD	IR^*^	IR^**^	IR^***^	nObs	nTrades
**panel A: BTC 1 d**									
BTC	132.41	78.98	83.23	2.96	1.68	2.67	1.19	1461	NA
MSE Long/Short	182.18	78.89	83.23	2.08	2.31	5.05	4.42	1461	4
MADL Long/Short	190.47	78.88	48.59	0.85	2.41	9.47	21.23	1461	156
MSE Long only	159.07	77.46	83.23	2.85	2.05	3.92	2.19	1461	2
MADL Long only	200.49	57.83	50.79	1.40	3.47	13.69	19.56	1461	78
**panel B: S&P500 1 d**									
S&P500	13.39	20.51	33.92	0.58	0.65	0.26	0.06	1005	NA
MSE Long/Short	0.44	20.53	28.61	2.04	0.02	0.00	0.00	1005	1014
MADL Long/Short	6.13	20.48	25.09	2.56	0.30	0.07	0.00	1005	168
MSE only	7.90	14.18	16.31	1.47	0.56	0.27	0.01	1005	507
MADL Long only	11.20	12.35	13.08	1.74	0.91	0.78	0.05	1005	84

Note: BTC and S&P500 stand for the benchmark strategies, i.e., Buy&Hold applied for BTC and S&P500 prices, respectively. MSE Long/Short and MADL Long/Short stand for the investment strategy with long and short signals from models optimized with MSE and MADL loss functions, respectively. MSE Long only and MADL Long only stand for the investment strategy with long only signals from models optimized with MSE and MADL loss function. The table presents the results in the period between 1 January 2017 and 31 December 2020 for daily frequency. The hyperparameters of LSTM model for the the base case scenario were set as it was described in Table 1.

**Table 4 sensors-22-00917-t004:** Performance metrics for investment strategies on BTC and S&P500 in the base case scenario, compared with benchmarks.

	aRC	aSD	MD	MLD	IR^*^	IR^**^	IR^***^	nObs	nTrades
**panel A: BTC 1 d**									
BTC	132.41	78.98	83.23	2.96	1.68	2.67	1.19	1461	NA
Long/Short	190.47	78.88	48.59	0.85	2.41	9.47	21.23	1461	156
Long only	200.49	57.83	50.79	1.40	3.47	13.69	19.56	1461	78
**panel B: BTC 1 h**									
BTC	134.66	87.01	83.94	2.96	1.55	2.48	1.13	35,063	NA
Long/Short	37.28	87.01	67.63	1.47	0.43	0.24	0.06	35,063	6398
Long only	115.62	62.77	56.33	1.40	1.84	3.78	3.11	35,063	3200
**panel C: BTC 15 min**									
BTC	134.93	89.35	84.01	2.96	1.51	2.43	1.11	140,255	NA
Long/Short	−44.04	89.35	96.51	2.13	−0.49	−0.22	−0.05	140,255	27,100
Long only	40.39	62.73	73.88	2.69	0.64	0.35	0.05	140,255	13,550
**panel D: S&P500 1 d**									
S&P500	13.44	20.47	33.97	0.58	0.66	0.26	0.06	1005	NA
Long/Short	6.13	20.48	25.09	2.56	0.30	0.07	0.00	1005	168
Long only	11.20	12.35	13.08	1.74	0.91	0.78	0.05	1005	84
**panel E: S&P500 1 h**									
S&P500	12.50	18.09	34.77	0.64	0.69	0.25	0.05	7041	NA
Long/Short	−2.69	18.09	48.04	1.51	−0.15	−0.01	0.00	7041	1554
Long only	5.54	12.39	19.50	1.51	0.45	0.13	0.00	7041	777
**panel F: S&P500 15 min**									
S&P500	13.56	19.08	35.34	0.59	0.71	0.27	0.06	26,155	NA
Long/Short	15.63	19.08	16.70	1.19	0.82	0.77	0.10	26,155	6444
Long only	15.74	12.87	8.25	0.97	1.22	2.33	0.38	26,155	3222

Note: BTC and S&P500 stand for the benchmark strategies, i.e., Buy&Hold applied for BTC and S&P500 prices, respectively. Long/Short stands for the investment strategy with long and short signals. Long only stands for the investment strategy with long only signals. The table presents the results in the period between 1 January 2017 and 31 December 2020 for daily frequency. The hyperparameters of LSTM model for the the base case scenario were set as it was described in Table 1.

**Table 5 sensors-22-00917-t005:** Results of regressions for returns: Long/Short and Long only strategies on BTC and S&P500 for three different frequencies.

	Alpha	Std. Err.	*t*	pv	Beta	Std. Err.	*t*	pv
**panel A: BTC 1 d vs.**								
Long/Short	0.0028	0.0011	2.5805	0.0100 **	0.0613	0.0260	2.3596	0.0184 *
Long only	0.0018	0.0005	3.3295	0.0009 ***	0.5258	0.0130	40.3095	0.0000 ***
**panel B: BTC 1 h vs.**								
Long/Short	0.0000	0.0000	0.6506	0.5153	0.0401	0.0053	7.5261	0.0000 ***
Long only	0.0000	0.0000	1.4969	0.1344	0.5190	0.0027	194.5037	0.0000 ***
**panel C: BTC 15 min vs.**								
Long/Short	0.0000	0.0000	−1.2728	0.2031	−0.0126	0.0027	−4.7162	0.0000 ***
Long only	0.0000	0.0000	−0.3720	0.7099	0.4944	0.0013	370.3576	0.0000 ***
**panel D: S&P500 1 d vs.**								
Long/Short	0.0004	0.0004	0.9501	0.3423	−0.2705	0.0302	−8.9672	0.0000 ***
Long only	0.0002	0.0002	1.2270	0.2201	0.3598	0.0152	23.7180	0.0000 ***
**panel E: S&P500 1 h vs.**								
Long/Short	0.0000	0.0001	−0.2379	0.8120	−0.0551	0.0119	−4.6340	0.0000 ***
Long only	0.0000	0.0000	−0.0377	0.9699	0.4720	0.0060	79.3050	0.0000 ***
**panel F: S&P500 15 min vs.**								
Long/Short	0.0000	0.0000	1.6450	0.1000 *	−0.0874	0.0061	−14.2609	0.0000 ***
Long only	0.0000	0.0000	1.8525	0.0640 *	0.4537	0.0031	147.3734	0.0000 ***

Note: BTC and S&P500 stand for the benchmark strategies, i.e., Buy&Hold applied for BTC and S&P500 prices, respectively. Long/Short stands for the investment strategy with long and short signals. Long only stands for the investment strategy with long only signals. The table presents the results of regressions in the form of: $R_{t}=\alpha +\beta R_{t}^{*}+\varepsilon_{t}$, where $R_{t}$ is the return for tested strategy in period *t* and $R_{t}^{*}$ is the return in of BTC or S&P500 strategies. Regressions were calculated in the period between 1 January 2017 and 31 December 2020. The hyperparameters of LSTM model for the the base case scenario were set as it was described in Table 1. Asterisks *, ** and *** denote statistical significance at the 10%, 1% and 0.1%, respectively.

**Table 6 sensors-22-00917-t006:** Sensitivity analysis for 1 h data with regard to the dropout rate.

	aRC	aSD	MD	MLD	IR^*^	IR^**^	IR^***^	nObs	nTrades
**panel A: BTC LS**									
BTC	134.66	87.01	83.94	2.96	1.55	2.48	1.13	35,063	NA
dr001	29.13	87.01	71.82	2.91	0.33	0.14	0.01	35,063	6434
dr002	37.28	87.01	67.63	1.47	0.43	0.24	0.06	35,063	6398
dr004	58.26	86.97	62.96	2.96	0.67	0.62	0.12	35,063	6534
**panel B: BTC LO**									
BTC	134.66	87.01	83.94	2.96	1.55	2.48	1.13	35,063	NA
dr001	111.01	61.28	51.13	1.41	1.81	3.93	3.10	35,063	3219
dr002	115.62	62.77	56.33	1.40	1.84	3.78	3.11	35,063	3200
dr004	134.11	62.38	56.15	2.41	2.15	5.13	2.86	35,063	3263
**panel C: S&P500 LS**									
S&P500	12.50	18.09	34.77	0.64	0.69	0.25	0.05	7041	NA
dr001	−0.45	18.09	27.54	1.76	−0.02	0.00	0.00	7041	1574
dr002	−2.69	18.09	48.04	1.51	−0.15	−0.01	0.00	7041	1554
dr004	−1.43	18.09	45.17	1.51	−0.08	0.00	0.00	7041	1614
**panel D: S&P500 LO**									
S&P500	12.50	18.09	34.77	0.64	0.69	0.25	0.05	7041	NA
dr001	6.78	12.19	14.10	0.96	0.56	0.27	0.02	7041	787
dr002	5.54	12.39	19.50	1.51	0.45	0.13	0.00	7041	777
dr004	6.32	11.65	26.61	1.03	0.54	0.13	0.01	7041	807

Note: BTC and S&P500 stand for the benchmark strategies, i.e., Buy&Hold applied for BTC and S&P500 prices, respectively. LS stands for the investment strategy with long and short signals. LO stands for the investment strategy with long only signals. dr001, dr002, dr004 is the abbreviation for dropout rates equal to 1%, 2%, and 4%. The table presents the results in the period between 1 January 2017 and 31 December 2020 for daily frequency. The hyperparameters of LSTM model for the base case scenario were set as it was described in Table 1.

**Table 7 sensors-22-00917-t007:** Sensitivity analysis for 1 h data with regard to the sequence length.

	aRC	aSD	MD	MLD	IR^*^	IR^**^	IR^***^	nObs	nTrades
**panel A: BTC LS**									
BTC	134.66	87.01	83.94	2.96	1.55	2.48	1.13	35,063	NA
seq10	−60.61	87.01	98.67	3.94	−0.70	−0.43	−0.07	35,063	12206
seq20	37.28	87.01	67.63	1.47	0.43	0.24	0.06	35,063	6398
seq40	19.49	87.02	97.77	2.93	0.22	0.04	0.00	35,063	4616
**panel B: BTC LO**									
BTC	134.66	87.01	83.94	2.96	1.55	2.48	1.13	35,063	NA
seq10	15.32	63.02	76.53	2.87	0.24	0.05	0.00	35,063	6106
seq20	115.62	62.77	56.33	1.40	1.84	3.78	3.11	35,063	3200
seq40	105.60	59.14	83.54	2.93	1.79	2.26	0.81	35,063	2308
**panel C: S&P500 LS**									
S&P500	12.50	18.09	34.77	0.64	0.69	0.25	0.05	7041	NA
seq07	6.02	18.09	20.34	1.97	0.33	0.10	0.00	7041	2922
seq14	−2.69	18.09	48.04	1.51	−0.15	−0.01	0.00	7041	1554
seq28	2.65	18.09	34.83	0.86	0.15	0.01	0.00	7041	912
**panel D: S&P500 LO**									
S&P500	12.50	18.09	34.77	0.64	0.69	0.25	0.05	7041	NA
seq07	10.14	12.57	12.65	0.91	0.81	0.65	0.07	7041	1461
seq14	5.54	12.39	19.50	1.51	0.45	0.13	0.00	7041	777
seq28	8.57	11.01	19.53	0.90	0.78	0.34	0.03	7041	456

Note: BTC and S&P500 stand for the benchmark strategies, i.e., Buy&Hold applied for BTC and S&P500 prices, respectively. LS stands for the investment strategy with long and short signals. LO stands for the investment strategy with long only signals. seq10, seq20, seq40, seq07, seq14, seq28 is the abbreviation for sequence lengths equal to 10, 20, 40, 7, 14, and 28. The table presents the results in the period between 1 January 2017 and 31 December 2020 for daily frequency. The hyperparameters of LSTM model for the base case scenario were set as it was described in Table 1.

**Table 8 sensors-22-00917-t008:** Sensitivity analysis for 1 h data with regard to the train set length.

	aRC	aSD	MD	MLD	IR^*^	IR^**^	IR^***^	nObs	nTrades
**panel A: BTC LS**									
BTC	134.66	87.01	83.94	2.96	1.55	2.48	1.13	35,063	NA
tr0685	−40.19	87.02	94.91	2.98	−0.46	−0.20	−0.03	35,063	6370
tr1371	37.28	87.01	67.63	1.47	0.43	0.24	0.06	35,063	6398
tr2742	8.35	87.00	88.78	1.46	0.10	0.01	0.00	35,063	6178
**panel B: BTC LO**									
BTC	134.66	87.01	83.94	2.96	1.55	2.48	1.13	35,063	NA
tr0685	43.78	60.89	73.21	2.99	0.72	0.43	0.06	35,063	3185
tr1371	115.62	62.77	56.33	1.40	1.84	3.78	3.11	35,063	3200
tr2742	93.18	60.37	69.59	1.48	1.54	2.07	1.30	35,063	3091
**panel C: S&P500 LS**									
S&P500	12.50	18.09	34.77	0.64	0.69	0.25	0.05	7041	NA
tr0474	3.03	18.09	32.91	1.89	0.17	0.02	0.00	7041	1516
tr0948	−2.69	18.09	48.04	1.51	−0.15	−0.01	0.00	7041	1554
tr1896	8.04	18.09	26.19	0.44	0.44	0.14	0.02	7041	1474
**panel D: S&P500 LO**									
S&P500	12.50	18.09	34.77	0.64	0.69	0.25	0.05	7041	NA
tr0474	8.67	11.90	27.25	1.02	0.73	0.23	0.02	7041	758
tr0948	5.54	12.39	19.50	1.51	0.45	0.13	0.00	7041	777
tr1896	11.44	10.63	13.81	0.38	1.08	0.89	0.27	7041	737

Note: BTC and S&P500 stand for the benchmark strategies, i.e., Buy&Hold applied for BTC and S&P500 prices, respectively. LS stands for the investment strategy with long and short signals. LO stands for the investment strategy with long only signals. seq10, seq20, seq40, seq07, seq14, seq28 is the abbreviation for sequence lengths equal to 10, 20, 40, 7, 14, and 28. The table presents the results in the period between 1 January 2017 and 31 December 2020 for daily frequency. The hyperparameters of LSTM model for the base case scenario were set as it was described in Table 1.

**Table 9 sensors-22-00917-t009:** Sensitivity analysis for 1 h data with regard to the batch size.

	aRC	aSD	MD	MLD	IR^*^	IR^**^	IR^***^	nObs	nTrades
**panel A: BTC LS**									
BTC	134.66	87.01	83.94	2.96	1.55	2.48	1.13	35,063	NA
bs040	6.67	87.02	76.87	3.33	0.08	0.01	0.00	35,063	6160
bs080	37.28	87.01	67.63	1.47	0.43	0.24	0.06	35,063	6398
bs160	54.61	87.02	71.63	2.02	0.63	0.48	0.13	35,063	6436
**panel B: BTC LO**									
BTC	134.66	87.01	83.94	2.96	1.55	2.48	1.13	35,063	NA
bs040	95.33	57.81	50.23	1.35	1.65	3.13	2.21	35,063	3081
bs080	115.62	62.77	56.33	1.40	1.84	3.78	3.11	35,063	3200
bs160	128.70	62.61	46.02	1.37	2.06	5.75	5.41	35,063	3218
**panel C: S&P500 LS**									
S&P500	12.50	18.09	34.77	0.64	0.69	0.25	0.05	7041	NA
bs040	0.76	18.09	32.77	2.17	0.04	0.00	0.00	7041	1582
bs080	−2.69	18.09	48.04	1.51	−0.15	−0.01	0.00	7041	1554
bs160	−5.83	18.09	42.30	1.51	−0.32	−0.04	0.00	7041	1562
**panel D: S&P500 LO**									
S&P500	12.50	18.09	34.77	0.64	0.69	0.25	0.05	7041	NA
bs040	7.53	10.77	12.64	1.20	0.70	0.42	0.03	7041	791
bs080	5.54	12.39	19.50	1.51	0.45	0.13	0.00	7041	777
bs160	3.95	11.43	21.45	0.90	0.35	0.06	0.00	7041	781

Note: BTC and S&P500 stand for the benchmark strategies, i.e., Buy&Hold applied for BTC and S&P500 prices, respectively. LS stands for the investment strategy with long and short signals. LO stands for the investment strategy with long only signals. bs040, bs080, bs160 is the abbreviation for batch sizes equal to 40, 80 and 160. The table presents the results in the period between 1 January 2017 and 31 December 2020 for daily frequency. The hyperparameters of LSTM model for the base case scenario were set as it was described in Table 1.

**Table 10 sensors-22-00917-t010:** Performance metrics for the combined model fir different frequencies and different assets.

	aRC	aSD	MD	MLD	IR^*^	IR^**^	IR^***^	nObs	nTrades
**panel A: BTC combined frequencies**									
BTC	134.93	89.35	84.01	2.96	1.51	2.43	1.11	140,255	NA
Long/Short	7.70	54.02	62.07	2.90	0.14	0.02	0.00	140,255	16902
Long only	81.89	52.65	49.55	1.47	1.56	2.57	1.43	140,255	4048
**panel B: S&P500 combined frequencies**									
S&P500	13.56	19.08	35.34	0.59	0.71	0.27	0.06	26,155	NA
Long/Short	5.99	14.06	21.27	0.76	0.43	0.12	0.01	26,155	4262
Long only	10.61	10.88	11.74	0.42	0.98	0.88	0.22	26,155	1134
**panel C: combined assets, RB = 3M, weights 10/90**									
BTC/SP500	31.72	21.57	35.88	1.51	1.47	1.30	0.27	140,255	NA
Long/Short	7.22	13.89	18.58	1.21	0.52	0.20	0.01	140,255	NA
Long only	19.62	11.75	12.43	0.55	1.67	2.64	0.94	140,255	NA
**panel D: combined assets, RB = 3M, weights 20/80**									
BTC/SP500	47.49	28.12	39.07	1.52	1.69	2.05	0.64	140,255	NA
Long/Short	7.80	15.84	18.60	1.16	0.49	0.21	0.01	140,255	NA
Long only	27.41	15.01	14.57	1.20	1.83	3.43	0.79	140,255	NA
**panel E: combined assets, RB = 6M, weights 10/90**									
BTC/SP500	35.51	22.96	35.88	1.51	1.55	1.53	0.36	140,255	NA
Long/Short	7.37	14.05	19.00	1.21	0.52	0.20	0.01	140,255	NA
Long only	21.37	12.19	12.43	0.55	1.75	3.01	1.18	140,255	NA
**panel F: combined assets, RB = 6M, weights 20/80**									
BTC/SP500	53.76	30.17	41.24	1.52	1.78	2.32	0.82	140,255	NA
Long/Short	8.05	16.21	17.70	1.16	0.50	0.23	0.02	140,255	NA
Long only	30.27	15.85	14.07	1.20	1.91	4.11	1.04	140,255	NA

Note: BTC and S&P500 stand for the benchmark strategies, i.e., Buy&Hold applied for BTC and S&P500 prices, respectively. Long/Short stands for the investment strategy with long and short signals. Long stands for the investment strategy with long only signals. The table presents the results in the period between 1 January 2017 and 31 December 2020 for daily frequency. The hyperparameters of LSTM model for the base case scenario were set as it was described in Table 1.

## Data Availability

All relevant data along with the sources are presented in the paper.

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
