# Peer review of "LSTM in Algorithmic Investment Strategies on BTC and S&P500 Index"

_sensors, 2022, doi:10.3390/s22030917_

Round 1

Reviewer 1 Report

The authors didn’t consider all the comments and suggestions 
The authors should take attention to all comments and suggestions 

the authors should give a real attention to all comments by reviewer which included in the peer review report 

Author Response

Attached please find our detailed response to your review in the separate .pdf file.

Reviewer 2 Report

Definition of MADE (8) is technically correct now. It is a loss function but why call it an "error" function as is represented by "E" in MADE. What is defined by equation (8) should be called "mean absolute directional return (or loss)" not "mean absolute directional error".

Author Response

(The authors gave the same response as above.)

Reviewer 3 Report

This paper analyzes deep-learning based algorithmic investment strategies for Bitcoin and stock markets. I selected this paper while not being an expert in the field due to some curiosity about the results. The paper seems to confirm the intuitive idea that while algorithmic investment can help to increase profits and reduce volatility, it is not silver bullet.

The authors formulate five research hypothesis, develop their approaches using a standard LSTM setup, analyze sensitivity of parameters and so on, and present results on historical market data. They are able to reject most (3 of 5) of the hypotheses, with the two remaining ones remaining open.

As this version is a revision, the presentation and language is quite good and I have no major comments on this.

The main thing I noticed about the background is that, as a non-expert, the amount of papers with bad methodology in this field seem to be astounding. Good job on categorizing the different problems with them!

My main problem with the present article is related to the claim about the superiority of their new loss function and the statement "Moreover, the proper loss function is crucial in the model estimation process." As far as I could tell, the paper does not substantiate these claims, or discuss how the loss function results in better performance, in the metrics that investors care about, such as Annualized Return Compounded, annualized SD, or the max drawdown.

A major revision request: The authors should either add an experimental comparison between their loss function and the ones from the related works, or reference a peer-reviewed article which already includes this comparison. In the second case, it should be made clear (in the Abstract, Intro etc.) that the loss function is not a novel contribution of this work.

As a minor note, I suggest that the authors reconsider the title of the paper in order to make it shorter / easier to read and more attractive. For one simple example, articles in titles are not required by the English grammar, so you can drop the word "The" in the title.

Author Response

(The authors gave the same response as above.)

Round 2

Reviewer 3 Report

Thanks you for the revised version, with the new Table 3 my main comment is addressed.

This manuscript is a resubmission of an earlier submission. The following is a list of the peer review reports and author responses from that submission.

Round 1

Reviewer 1 Report

See the attached report.

Author Response

The detailed answers to the Reviewer are placed in the attached .pdf file.

Reviewer 2 Report

All the comments and suggestions on the current manuscript are available in attached peer review report.

Author Response

(The authors gave the same response as above.)

Reviewer 3 Report

The authors have used LSTM networks to forecast the value of the BTC and SP 500 index on the data
2 from 2013 to the end of 2020 on the following frequencies: daily, 1h and 15min data. 

I like the paper. The best part of the paper is empirical results. It gives a general overview of the performance of the proposed methods using various factors/metrics. 

To complete the work, I would highly recommend to add a statistical test to see if the outperformance is statistically significant. 

There are some tests, but the one that fits very well for this work is Hassani and Silava KM test for comparison of several forecast outputs. 

Here is the link for the test and available R code:

https://www.mdpi.com/2225-1146/3/3/590

This will enhance the quality of the work. 

Author Response

(The authors gave the same response as above.)

Reviewer 4 Report

This manuscript does not give a clear description of the scientific problem it aims to investigate and address. Does the manuscript intend to compare the LSTM method with other machine learning methods, or use the LSTM method to compare the performance of AIS's in different financial datasets? Without this clarification, it is very difficult to assess the manuscript's contribution.

Architecture of the LSTM network described by equations (1) to (7) are very confusing. Specifically, not all variables or terms there are defined or specified; and it is not clear how these variables relate to any features in BTC and SP500 index data. Equations (3) and (4) are different but for describing the same it.  

MADE defined in equation (8) does not seem to make sense, as it involves only Rp,i and Rt,i but not their difference or forecast error Rp,i − Rt,i. MADE should equal 0 if the forecast error equals 0.

There are many English grammar errors and misprints, making the manuscript very difficult to read.

Author Response

(The authors gave the same response as above.)

Round 2

Reviewer 1 Report

See attached file.

Reviewer 2 Report

The literature review should be improved and presented a new fresh related studies.

The results should be presented clearly.

The conclusions should supported the results.

Reviewer 4 Report

There are still a lot of English grammar errors, typos and misprints. Examples: 1. It should be "its" not "it's" in line 23.

2. In line 214, it should be "it" not "it".

3. In lines 301 to 314, "chapter" should be replaced by "section".

The authors need to thoroughly proof-read the manuscript to correct these errors/typos as much as possible.

The proposed MADE defined by (8) is not a loss or error function as claimed in the manuscript. It is actually a directional fitted or predicted value function giving the next period returns value. MADE is not computable in real-world cases when Rt,i has not been observed. The authors need to provide some numerical results of MADE using additional simulation study.